# DNA Methylation Profile Changes in CpG Islands of Ethylene-Signaling Genes Regulated by Melatonin Were Involved in Alleviating Chilling Injury of Postharvest Tomato Fruit

**DOI:** 10.3390/ijms26136170

**Published:** 2025-06-26

**Authors:** Jingrui Yan, Shuangshuang Shan, Jiangkuo Li, Zhengke Zhang, Jiali Yang, Wanli Zhang, Hongmiao Song, Xiangbin Xu, Wenhui Duan

**Affiliations:** 1College of Food Science and Engineering, Hainan University, Haikou 570228, China; yanjingrui0316@163.com (J.Y.); shanshuangs@163.com (S.S.); zhangzhengke@hainanu.edu.cn (Z.Z.); jialiyang@hainanu.edu.cn (J.Y.); zwl@hainanu.edu.cn (W.Z.); hmsongibcas@126.com (H.S.); 2Tianjin Key Laboratory of Postharvest Physiology and Storage of Agricultural Products, National Engineering and Technology Research Center for Preservation of Agricultural Products, Tianjin 300384, China; lijkuo@sina.com

**Keywords:** chilling injury, DNA methylation, ethylene signaling, melatonin, tomato

## Abstract

Melatonin (MT) has been reported to alleviate chilling injury (CI) in postharvest tomato fruit during low-temperature storage. In the present study, the DNA methylation profile changes in the CpG islands of ethylene signaling genes regulated by MT in postharvest tomato fruit during low-temperature storage were detected. The MT treatment increased the content of total soluble solids (TSS) and enhanced the ethylene production of tomato fruit. Moreover, it decreased titratable acidity (TA) content, inhibited the activity of polygalacturonase (PG), and kept the firmness of tomato fruit under low-temperature storage. In the MT-treated tomato fruit, significant changes in DNA methylation of CpG island of *SlACS10*, *LeCTR1*, *LeEIN3*, *SlERF-A1*, and *LeERT10* genes were induced; the expression of *LeCTR1* was inhibited; and the expression of *SlACS10*, *LeEIN3*, and *SlERF-A1* genes was increased, by which the ethylene signaling might be influenced and the CI was alleviated. The present results provide evidence that the CI of postharvest tomato fruit alleviated by MT might be related to the changes in DNA methylation of ethylene-signaling genes.

## 1. Introduction

Low-temperature storage is a postharvest storage technology that is widely used to delay postharvest senescence of horticultural products [1]. Low-temperature storage could preserve the nutritional quality of postharvest fruit and vegetables, and delay their lifespan [2]. However, many fruit and vegetables are sensitive to low temperature and suffer from different physiological diseases due to chilling injury (CI). Tomato is a cold-sensitive fruit that originated in the tropical region [3]. The CI symptoms of tomato fruit are diverse, including tissue browning, uneven maturity, water-soaked areas, pitting of the skin, and increased susceptibility to disease [4,5]. These harmful changes reduce quality and consumer acceptance, and result in significant economic losses. Recently, many strategies, including far-red light [6], applications of exogenous oxalic acid [7], salicylates and jasmonates treatment [8], hot air treatment [9], ultraviolet irradiation [10], and methyl jasmonate-induced [11], have been presented. These approaches effectively improve the chilling tolerance of postharvest tomato fruit. In addition, exogenous application of melatonin (*N*-acetyl-5-methoxytryptamine, MT) also provides sufficient intracellular ATP by enhancing Ca-ATPase, H-ATPase, succinate dehydrogenase, and cytochrome c oxidase enzymes’ activity; preserves membrane fluidity and integrity by improving the expression of FAD3 and FAD7 genes; and alleviates the chilling damage of tomato fruit [12].

MT, as an important bioactive compound, was first extracted from the pineal glands of mammalians, and subsequently, it was also discovered to come from vascular plants as well [13,14]. Its precursor, tryptophan, was converted to serotonin via 5-hydroxytryptophan, and then serotonin was acetylated to *N*-acetylserotonin by the enzyme serotonin-*N*-acetyltransferase and last methylated to yield MT by hydroxyindole-O-methyltransferase [15]. MT was considered to be the effective antioxidant and powerful free-radical scavenger; meanwhile, it was also a plant hormone widely presenting in various tissues, which are involved in a variety of physiological actions, including photosynthesis, seed germination, plant growth and development, and fruit ripening and senescence, as well as playing a role against multiple biotic and abiotic stresses [16,17,18]. It counteracted cell accumulation of H_2_O_2_; delayed leaf senescence of rice [19]; deferred fruit senescence of pear and sweet cherry [20,21]; and enhanced chilling tolerance of peach, pomegranate, and litchi fruit [22,23,24].

DNA methylation is considered an epigenetic marker of transcriptional gene silencing that is usually associated with inactive transcription in mammals and plants [25]. It plays a key role in various biological processes, such as gene regulation, genome stability, genomic immunity, development, and responses to the stress conditions [26]. DNA methylation in eukaryotes refers to the addition of a methyl group of cytosine carbon 5 [27]. In plants, DNA methylation usually occurs in all sequences of cytosine bases: asymmetric CHH context (where H is any nucleotide except G), the CHG and symmetric CG contexts [28]. In addition, the dynamics of DNA methylation and demethylation determined the levels of DNA methylation [26]. Recent studies have shown that dynamic changes in DNA methylation played an important role in the ripening of fleshy fruit. In tomato fruit, DNA methylation affected m6A modification by regulating the expression of N6-methyladenosine (m6A) demethylase gene, and m6A demethylase feedback regulated DNA methylation, thus regulated fruit ripening [29]. Differential DNA methylation of the MYB10 promoter in apples altered the accumulation of MYB10 transcripts, which in turn altered the transcription and pigment accumulation of anthocyanin structural genes [30]. During the ripening of the non-climacteric fruit the strawberry, the reduction in RdDM (RNA-directed DNA methylation pathway)-mediated methylation versus the addition in DNA demethylation indicated that the regulation of RdDM to the genetic program contributed to the ripening of strawberry fruit [31]. In addition, the increase in global DNA methylation promoted the development and ripening of sweet orange fruit [32]. The DNA demethylase gene DNA demethylases 2 mutations caused DNA hypermethylation during tomato fruit ripening [26]. To date, the epigenetic mechanism of MT involved in alleviating chilling damage to fruit is still unclear. In the present study, the DNA methylation profile changes in CpG islands of ethylene-signaling genes regulated by MT in tomato fruit were detected, and the epigenetic mechanism of MT alleviating CI of postharvest tomato fruit was analyzed.

## 2. Results

### 2.1. Effects of MT on CI Symptoms, Cellular Ultrastructure, and CI Index

After 15 d of low-temperature storage, CI symptoms were observed in the control fruit, while MT treatment significantly alleviated the occurrence of CI (Figure 1A–C). At low-temperature storage for 15 d, visible pitting spots were observed on the surface of the control tomato fruit, and the inner tissue of the fruit pericarp suffered from CI. No CI symptoms occurred in MT-treated fruit at 15 d. At low-temperature storage for 18 d, the CI symptoms of the surface of the control fruit covered more than 25%, and the MT-treated fruit showed mild CI symptoms (Figure 1A,B). The TEM results showed that the morphological structure of epidermal cells in tomato fruit change from low-temperature storage (Figure 1C). In the control fruit, the cell edge shrank, and the symptoms of nuclear karyorrhexis occurred after 15 d; apoptotic cells further appeared at 18 d. In MT-treated fruit, the nucleus and vacuole displayed a relatively complete state at 15 d, and the slight cell shrinkage, nuclear condensation, and vacuole compartmentalization were observed at 18 d. The CI index of tomato fruit treated with MT significantly decreased (Figure 1D). In control fruit, the CI index was 4.2 and 13.3% at 15 and 18 d, respectively. In MT-treated fruit, the CI index was only 5.8% at 18 d.

### 2.2. Effects of MT on Firmness, Ethylene Production, TSS, TA, and PG Activity

The firmness of MT-treated fruit decreased to 9.4 N at 18 d (Figure 1E). In control fruit, the firmness was 7.5 N at 18 d. Compared with the firmness of the control fruit, the MT treatment maintained a higher level of firmness in tomato fruit. The ethylene production was increased in both control and MT-treated fruit during the whole low-temperature storage (Figure 1F). In control fruit, the ethylene production levels were below 104.13 ng kg^−1^ s^−1^. In the MT-treated fruit, the ethylene production was significantly higher than that in the control fruit from 9 to 15 d, and it reached a peak value of 130.07 ng kg^−1^ s^−1^ at 18 d. The content of TSS in MT-treated tomato fruit was higher than that in the control fruit, and the difference between the two groups was significant at 6, 15, and 18 d (Figure 1G). The TA content in the control fruit and MT-treated fruit fluctuated between 0.73% and 0.63% (Figure 1H). Compared with that in the control fruit, the TA content in the MT-treated fruit slightly reduced. As shown in the Figure 1I, the activity of PG in both control and MT-treated fruit was inhibited under low-temperature storage, and it was significantly reduced by MT treatment.

### 2.3. Electronic Nose Response to Fruit Aroma

LDA showed that the total variance was 84.03% (Figure 2A). LD1 and LD2 explained 58.09 and 25.94% of the total variance, respectively. Compared with the fruit in the early stage of storage, both groups showed a clear downward displacement in negative direction on the ordinate axis (LD2) at 18 d. The total variance was 98.13% in the results of the PCA (Figure 2B). PC1, the first principal component, accounted for 82.56% of the total variation, while PC2 accounted for 15.57% of the total variation. PCA showed that there was a certain trend of change in the abscissa (PC1) of both groups. Compared with that shown at 15 d, the two groups showed positive progress in function 1 (PC1) at 18 d. The separation of the control fruit and the MT-treated fruit showed that the volatile substances in the MT-treated fruit changed at 18 d (Figure 2B). Compared with LDA, PCA could detect the differences of tomato-fruit volatile aroma between the MT-treated group and the control group more clearly.

### 2.4. Effects of MT on Phenolic Compounds

The 11 kinds of phenolic compounds were identified in tomato fruit, and MT treatment maintained higher phenolic compounds (Figure 2C–D). Among them, phenylalanine and rutoside, with an initial content of 44.421 and 42.539 mg kg^−1^, were the two most abundant phenolic compounds in tomato fruit. The content of caffeic acid and rutoside in MT-treated fruit was significantly higher (*p* < 0.05) than in the control at 9 and 18 d. The content of caffeic acid in MT-treated fruit increased from 0.145 to 0.267 mg kg^−1^ during storage from 9 to 18 d, while in the control, it decreased from 0.101 to 0.045 mg kg^−1^. The rutoside content in the control and MT-treated fruit reached its peak at 9 d, i.e., 59.887 and 67.513 mg kg^−1^, respectively. The content of ferulic acid, narcissin, and hesperidin in MT-treated fruit was 1.28, 1.25, and 1.47 times higher, respectively, than that of the control at 18 d.

### 2.5. Effects of MT on Methylase and Demethylase Activity

Compared with that in the control fruit, the activity of methylase significantly decreased in the MT-treated fruit (Figure 3A). In the control fruit, the activity of methylase was increased first and then decreased, and it reached a peak value of 20.20 U kg^−1^ at 9 d. The activity of methylase in the MT-treated fruit was 6.59 U kg^−1^ at 9 d (Figure 3A).

The activity of demethylase in MT-treated fruit was higher than that in the control fruit from 9 to 18 d (Figure 3B). In the control fruit, the demethylase activities were 5.67, 4.87, and 4.76 U kg^−1^ at 9, 15, and 18 d, respectively. In the MT-treated fruit, the demethylase activities were 6.64, 8.84, and 8.79 U kg^−1^ at 9, 15, and 18 d, respectively (Figure 3B).

### 2.6. Effects of MT on the Levels of Genes’ Expression and DNA Methylation

DNA methylation plays an important role in the regulation of gene expression. Low levels of gene expression are often associated with high levels of DNA methylation of the CpG-island regions of genes. To investigate the function of DNA methylation of CpG islands of ethylene-signaling genes regulated by MT on the CI of fruit, the levels of DNA methylation of CpG islands and expression of *SlACS10*, *LeCTR1*, *LeEIN3*, *SlERF-A1*, and *LeERT10* genes in fruit during low-temperature storage were analyzed.

MT treatment increased the expression level of *SlACS10* at 18 d (Figure 4A). The expression level of *SlACS10* in the control fruit and MT-treated fruit was 0.84 and 2.28 at 18 d, respectively. MT treatment decreased the level of DNA methylation of *SlACS10* at 18 d (Figure 4B). The DNA methylation level of CpG island of *SlACS10* in MT-treated fruit was decreased to 0 and 0.7% at 9 and 18 d, respectively (Figure 4B). However, the level of DNA methylation did not alter in the control fruit and remained at 1.1%. In the MT-treated fruit, the gene expression increased as the level of DNA methylation of CpG island of *SlACS10* was decreased at 18 d. Compared with that in the control fruit, MT treatment changed the distribution of DNA methylation sites of *SlACS10* in fruit at 18 d (Figure 4C). The expression levels of *LeCTR1* in the control fruit were 0.28 and 0.55 at 9 and 18 d, respectively (Figure 4D). In the MT-treated fruit, the expression levels of *LeCTR1* were 0.25 and 0.17 at 9 and 18 d, respectively. MT treatment decreased the expression level of *LeCTR1* compared with the control fruit. The DNA methylation changes in CpG island of *LeCTR1* were not detected in both of the control fruit and MT-treated fruit at 9 and 18 d (Figure 4E,F).

The expression levels of *LeEIN3* in the control fruit were 0.09 and 0.18 at 9 and 18 d, respectively (Figure 5A). In the MT-treated fruit, the expression levels of *LeEIN3* were 0.04 and 0.69 at 9 and 18 d, respectively. Compared with that in the control fruit, MT treatment increased the expression level of *LeEIN3* in fruit at 18 d. The DNA methylation levels of the CpG island of *LeEIN3* in the control fruit were 7.8 and 10% at 9 and 18 d, respectively (Figure 5B). In the MT-treated fruit, the DNA methylation levels of the CpG island of *LeEIN3* were 14.4 and 7.8% at 9 and 18 d, respectively. MT treatment decreased the DNA methylation level of the CpG island of *LeEIN3* at 18 d. Compared with that in the control fruit, MT treatment changed the distribution of DNA methylation sites of *LeEIN3* in fruit at 9 and 18 d (Figure 5C). The expression level of *SlERF-A1* increased in MT-treated fruit (Figure 5D). The expression levels of *SlERF-A1* in the control fruit were 0.55 and 0.50 at 9 and 18 d, respectively. In the MT-treated fruit, the expression levels of *SlERF-A1* were 1.03 and 0.51 at 9 and 18 d, respectively. Compared with that in the control fruit, MT treatment decreased the DNA methylation level of the CpG island of *SlERF-A1* in tomato fruit at 9 d (Figure 5E). The DNA methylation levels of the CpG island of *SlERF-A1* in control fruit were 0.9 and 0% at 9 and 18 d, respectively. In the MT-treated fruit, the DNA methylation level of the CpG island of *SlERF-A1* was 1.1% at 9 d and remained so at 18 d (Figure 5E,F).

The expression levels of *LeERT10* in control fruit were gradually decreased; the levels were 0.30 and 0.14 at 9 and 18 d, respectively (Figure 6A). In the MT-treated fruit, the expression levels of *LeERT10* were 0.16 and 0.87 at 9 and 18 d, respectively. Compared with that in the control fruit, MT treatment significantly increased the expression level of *LeERT10* in fruit at 18 d. The DNA methylation of CpG island of *LeERT10* in control fruit only occurred at 18 d, with a DNA methylation level of 0.8% (Figure 6B). On the contrary, the DNA methylation level of the CpG island of *LeERT10* in the MT-treated fruit was 0.4% at 9 d, and no CG sites were methylated at 18 d (Figure 6B,C).

## 3. Discussion

MT plays an important role in alleviating the CI of horticultural crops [22,33,34]. It also was applied to reduce CI, promote ripening, and improve quality of postharvest tomato fruit [12,35]. According to the present results, MT treatment alleviated the CI, retained the firmness and phenolic compounds, inhibited the activity of PG, increased TSS content and ethylene production, and decreased the TA content of postharvest tomato fruit (Figure 1 and Figure 2). Moreover, MT treatment significantly inhibited the activity of methylase (Figure 3A), and it promoted the activity of demethylase in tomato fruit during low-temperature storage (Figure 3B). Melatonin treatment alleviated CI symptoms and maintained the quality of postharvest tomato fruit, which possibly regulated methylase activity in order to change the DNA methylation profile of the CpG island of genes in ethylene biosynthesis and signaling (Figure 7).

Ethylene plays a positive role in regulating chilling tolerance of tomato fruit [36]. ACS is the main enzyme controlling the rate of ethylene biosynthesis [37]. At least 13 *ACS* genes have been identified in tomato fruit [38]. The production of a large amount of ethylene in tomato fruit was the reason for the increasing of *Le-ACS2* and *Le-ACS4* transcripts, indicating that *Le-ACS2* and *Le-ACS4* were positively regulated during ethylene biosynthesis [39]. In addition, antisense *SlACS2* blocked the ethylene biosynthesis and greatly reduced the chilling tolerance of tomato fruit during cold storage, thus indicating that the ethylene biosynthesis regulated by *SlACS2* has a positive effect on chilling tolerance of tomato fruit [36]. In the present results, the DNA methylation level of the CpG island of *SlACS10* in MT-treated fruit was lower than that in the control fruit, and the expression level of *SlACS10* significantly increased in MT-treated fruit compared to that in control fruit at 18 d (Figure 4A–C), and these effects might have been involved in ethylene biosynthesis and enhanced the chilling tolerance of tomato fruit during low-temperature storage. Interestingly, in the MT-treated fruit, both the DNA methylation and gene expression levels of *SlACS10* at 9 d were relatively lower than they were at 0 and 18 d (Figure 4A,B). In tomato fruit, the methylcytosine was often found to be at CG sites (71.60–72.30%), and the frequency of methylation in CHG and CHH sequences was 52.50–53.00% and 10.70–12.50%, respectively [40]. The DNA methylation of cytosines at CHG or CHH sites induced by MT might also be involved in affecting the expression of *SlACS10*.

CTR1, a negative regulator of ethylene signaling, functions upstream of the ethylene receptors [41,42]. In Arabidopsis, CTR1 blocks ethylene signaling by phosphorylating EIN2 protein [43]. The loss-of-function alleles of CTR1 indicated that the interaction of CTR1 and ETR1 was necessary for CTR1 to negatively regulate the ethylene signal [44]. Interestingly, in the present study, the MT treatment showed no effect on the CpG island of *LeCTR1* compared to the control fruit (Figure 4E,F). The DNA methylation level of the CpG island of *LeCTR1* was 0% in both the control and MT-treated fruit at 9 and 18 d. Still, the expression level of *LeCTR1* decreased at 9 and 18 d in MT-treated fruit, which might promote ethylene signal transduction and was involved in alleviating CI in tomato fruit. We speculated that the DNA methylation on cytosines at CHG or CHH sites induced by MT might affect the expression of *LeCTR1*.

EIN3 is a transcription factor that regulates the transcription of early ethylene-responsive genes in plants, and it is an important downstream component of the ethylene pathway [45]. The expression of the *EIN3* gene in tomato activated the ethylene conduction process, activated the expression of a series of ethylene-regulated target genes, and played a positive role in ethylene response [46]. In the MT-treated tomato fruit, the expression level of *LeEIN3* was much higher than that in the control fruit at 18 d (Figure 5A). The DNA methylation level of the CpG island of *LeEIN3* in tomato fruit was decreased by MT treatment at 18 d (Figure 5B,C). The present results suggested that MT treatment induced the DNA methylation changes and the expression of *LeEIN3*, and influenced the ethylene signaling of tomato fruit. Hence, the decreased DNA methylation level of the CpG island of *LeEIN3* in MT-treated fruit might have been involved in activating *LeEIN3* expression, by which ethylene signaling was promoted and CI in tomato fruit was alleviated.

Ethylene response factors (ERFs), one of the largest families of plant transcription factors, are the downstream element of the ethylene-signaling pathway [47]. ERFs are involved in the positive and negative regulation of ethylene response genes’ expression [48,49]. In addition, they play an important role in the cold response of fruit [50,51,52]. The *ERF* gene (*VaERF057*) might be a positive regulator of cold tolerance in grapevine, and its overexpression enhanced the cold tolerance of *Arabidopsis* [53]. The overexpression of *TERF2/LeERF2* improved the cold tolerance of tomato fruit by regulating ethylene signaling [51]. *SlERF-A1* is a member of the B3 group of ERF genes in tomato [54]. In the present study, MT treatment significantly increased the expression of *SlERF-A1* and decreased the DNA methylation level of CpG island of *SlERF-A1* in tomato fruit at 9 d (Figure 5D–F). The decreased DNA methylation level of the CpG island of *SlERF-A1* in MT-treated fruit might have been involved in activating *SlERF-A1* expression, by which it promoted ethylene signaling and was involved in alleviating fruit CI.

ERT genes were the clone series of Early Ripening Tomato (ERT) in tomato fruit, and the mRNAs of ERT in wild-type immature green fruit were present at a substantial level [55]. The mRNAs homologous to *LeERT10* in tomato fruit showed mature specific accumulation, and they reached the peak value at 3 to 5 d post-breaker stage [55]. In the present study, MT treatment significantly increased the expression level of *LeERT10* at 18 d (Figure 6A). The DNA methylation level of CpG island of *LeERT10* was reduced to 0% in the MT-treated tomato fruit at 18 d (Figure 6B,C). Thereby, the changes in the DNA methylation level of the CpG island of *LeERT10* induced by MT might have upregulated *LeERT10* expression and promoted the ethylene signal transduction, possibly serving as one of the reasons why the CI in tomato fruit was alleviated.

## 4. Materials and Methods

### 4.1. Fruit and Treatment

Cherry tomato (*Solanum lycopersicum* cv. Qianfu) fruit was harvested at the mature green stage in Tianjin, China, and transported to the laboratory immediately. The selected fruits were uniform in size, with bright coloration, without any blemishes or diseases, and without mechanical damage. All fruits were randomly divided into two batches. The first batch of fruit was immersed in a 500 µM MT solution at 25 °C for 10 min. Under the same conditions, the fruits of the other batch were dipped in distilled water and served as the control group. All fruit was air-dried for 30 min at room temperature. Each batch of fruit was divided into six groups of ten fruits, and each of the six groups was placed inside a clean polypropylene plastic box and sealed with a 0.02 mm thick low-density polyethylene bag [7]. Next, all fruit was stored at 0 °C with a relative humidity of 85 to 90%. Samples were taken from the low-temperature storage and equilibrated at room temperature (22 °C) for 2 h before biochemical analysis. Each treatment was replicated three times, and the experiment was repeated twice.

### 4.2. Histological Observations of Pericarp

The pericarp tissue at the equatorial part was cut into pieces with dimensions of 2 mm × 2 mm × 2 mm; after that, the pieces were immersed in 4% glutaraldehyde fixative for 2 h at room temperature and then stored at 4 °C overnight. The tissue was washed using phosphate buffer and then fixed with 1% osmium tetroxide for 4 h. The tissue, after dehydration, was embedded with Spurr embedding resin and was sliced. The cell ultrastructure of tomato tissue was observed by transmission electron microscopy (TEM) (JSM-IT700 HR, Tokyo, Japan) according to the method of Hou et al. [56].

### 4.3. CI Index

The CI index was assessed according to the method of Ding et al. (2002) [57], where no pitting = 0; pitting covering of the fruit surface < 25% = 1; pitting covering of the tomato surface between 25% and 50% = 2; pitting covering of the tomato surface between 50% and 75% = 3; and pitting covering of the tomato surface > 75% = 4. And the results were calculated using the method described by Zhao et al. (2009) [3]. The average range of CI index is calculated as follows: CI index (%) = {Σ[(CI level) × (Number of fruit at the CI level)]/(Total number of fruit) × 4} × 100%.

### 4.4. Firmness, Ethylene Production and Total Soluble Solids, Titratable Acidity, and Polygalacturonase Activity

Fruit firmness was assessed according to the method of Xu et al. (2012) [58], with some modification. It was determined on skin puncture strength of 10 intact fruits from each replicate, using a TA-XT plus texture analyzer (Stable Micro System Ltd., Godalming, Surrey, UK). Each fruit was measured on two paired sides with a probe with a diameter of 2 mm, depression of 6 mm, and speed of 2 mm s^−1^. The method described by Zhang et al. (2016) [59] was used to determine ethylene production, with some modification. Five of the fruits were sealed in 600 mL containers for 2 h, 1 mL of gas from each container was extracted by a gas-tight syringe, and then the gas sample ethylene was quantitatively analyzed by FID gas chromatograph (GC-2010, Shimadzu, Kyoto, Japan). The temperatures of the detector, injection port, and oven were 160, 150, and 60 °C. The ethylene production was expressed as ng kg^−1^ s^−1^.

Total soluble solids (TSS) and titratable acidity (TA) were determined via the juice of each sample fruit by the Pocket Brix-Acidity Meter (PAL-BXIACID F5, ATAGO, Tokyo, Japan). The fresh fruit tissue (20 g) was ground, and the juice was filtrated with gauze. The TSS was measured with 0.2 mL tomato juice sample by the Pocket Brix-Acidity Meter. The TA was measured with 0.2 mL tomato juice sample (which was diluted 50 times with distilled water) via the tomato mode of the Pocket Brix-Acidity Meter. According to the manufacturer’s instructions, the activity of polygalacturonase (PG) was detected by the plant PG ELISA kit (Jingkang Biological Engineering Co. Ltd., Shanghai, China) and measured with a microplate reader. The activity of PG was expressed with the fresh weight as units per kilogram (U kg^−1^).

### 4.5. Volatile Aroma Evaluation

According to the method of Defilippi et al. (2009) [60], the volatile aroma components of tomato fruit were determined by a portable electronic nose PEN3 (AIRSENSE, Schwerin, Mecklenburg-Vorpommern, Germany). Five fruits were placed in a sealed glass jar with a capacity of 0.6 L. The glass jar was closed and kept at room temperature (22 °C) for 10 min to allow for headspace equilibrium to be achieved. Then, the headspace sample gas was drawn into the sensor chamber through the inlet at a rate of 100 mL min^−1^. The procedure for the determination of aroma components via the electronic nose was as follows: sensor cleaning for 70 s, auto zero for 10 s, sample preparation for 5 s, sample test for 50 s, sample measurement interval for 1 s, and internal flow rate of 100 mL min^−1^.

### 4.6. Detection of Phenolic Compounds

The polyphenol compounds of tomato fruit were determined according to the method of Duan et al. [61], with minor modifications. A total of 4 g of powder of pericarp tissues was dipped in 4 mL of 70% (*v*/*v*) ethyl alcohol for 2 h, and then it was ultrasonically extracted for 30 min. The extractions were centrifuged at 10,000 rpm for 15 min at 25 °C, the supernatant was collected, and then a 4 mL aqueous solution of 70% (*v*/*v*) ethanol was added and ultrasonically extracted for 30 min again. The extractions were diluted with 70% (*v*/*v*) ethanol to 10 mL, and after the liquid was filtered through a 0.2 μm microporous filter membrane, the content was determined by UPLC-MS/MS (Waters ACQUITY UPLC I-Class/Xevo TQ-S IVD system, Milford, Massachusetts, USA). A total of 10 µL of extracts was injected on a Waters ACQUITY UPLC BEH C18 column (1.7 µm, 2.1 × 50 mm) to perform separations, and the column temperature was 35 °C. The mobile phase was composed of 0.25% (*v*/*v*) formic acid water and 0.25% (*v*/*v*) formic acid acetonitrile at a flow rate of 0.3 mL min^−1^. The PDA-scan wavelength was performed from 200 to 360 nm, and target compounds were detected by a multiple reaction monitoring (MRM) mode. The quantitative analysis of individual phenolic compounds in tomato fruit used an external standard method, and the results were expressed as mg kg^−1^.

### 4.7. Detection of Methylase and Demethylase Activity

The methylase and demethylase activity were detected, respectively, by Plant Methylase kit and Plant Demethylase kit (Gelatins Biotechnology Co. Ltd., Shanghai, China). According to the manufacturer’s instructions, the ground powder (1 g) of frozen tomato tissues was dissolved by 960 μL PBS in a 1.5 mL centrifuge tube and centrifuged at 3000 rpm for 20 min at 4 °C. The supernatant was used as the sample. Standards and samples were added to the plate wells and incubated for 30 min at 37 °C. The wells were washed with the 20-fold washing solution, and the conjugated antibody was added to the plate wells and incubated for 30 min at 37 °C. Then, the reaction was terminated by the addition of a sulfuric acid solution, and the color change was measured via spectrophotometry, at a wavelength of 450 nm. The concentration of methylase and demethylase activity were calculated using standard curves and expressed with U kg^−1^.

### 4.8. Expression Analysis of Genes Related to Ethylene Signaling

Total RNA was extracted from tomato tissues using TRIzol Reagent (Invitrogen, Carlsbad, California, USA). According to the method of Xu et al. (2013) [62], the relative expression levels of the five selected genes, namely *SlACS10*, *LeCTR1*, *LeEIN3*, *SlERF-A1*, and *LeERT10*, were measured simultaneously by quantitative real-time PCR (qRT-PCR). All reactions were performed in triplicate for each sample. GAPDH was used as an endogenous control, and the comparative Ct method (2^−ΔΔCt^) was adopted to calculate the expression data. The primers for qRT-PCR were designed with Primer 5.0. The primers used for qRT-PCR analysis are listed in Table 1.

### 4.9. Bisulfite Sequencing PCR and DNA Methylation Analysis

Total DNA was extracted from tomato tissues according to the method of Murray and Thompson (1980) [38]. The CpG islands of related genes were predicted by online software (http://www.bioinformatics.org/sms2/cpg_islands.html (accessed on 10 December 2024)). The details of the size and location of CpG island of genes are described in the Appendix A. Primers (Table 2) were used to amplify the target DNA regions in both control and treated tomato fruit. The PCR conditions were as follows: 95 °C for 5 min; 94 °C for 30 s, 50 °C for 30 s, and 72 °C for 30 s. The annealing temperature of the reaction was from 65 °C to a ‘touchdown’ at 55 °C by decreasing by 0.5 °C every cycle, followed by 30 cycles of 94 °C for 30 s, 50 °C for 30 s, and 72 °C for 30 s; finally, we kept it at 60 °C for 30 min. The PCR products were cloned into the vector, and more than 10 clones of each derivative were sequenced. The DNA methylation status of the sequences was aligned and analyzed via a BiQ analyzer (Version 0.7).

### 4.10. Statistical Analysis

All data were analyzed by one-way analysis of variance (ANOVA). Mean separations were performed by Duncans multiple range test. For electronic nose, sensor signals measured from 47 to 49 s are selected for electronic nose analysis, and the Win-Muster software of PEN3 electronic nose (AIRSENSE, Schwerin, Mecklenburg-Vorpommern, Germany) is used for linear discriminant analysis (LDA) and principal component analysis (PCA).

## 5. Conclusions

In summary, MT treatment induced changes in the DNA methylation level of the CpG island of *SlACS10*, *LeCTR1*, *LeEIN3*, *SlERF-A1*, and *LeERT10* genes to regulate the gene expression and enhance the ethylene production, which might play important roles in alleviating the CI and maintaining the quality of postharvest tomato fruit. Specifically, MT treatment effectively maintained the cell integrity, phenolic compounds, and volatile aroma of tomato fruit during low-temperature storage. These results offer a foundation and new strategies for future genetic modification via the epigenetic target sites of genes, and which contribute to alleviate economic and nutritional losses of postharvest tomato fruit by biotechnology to breed new cultivars in future studies. 

## Figures and Tables

**Figure 1 ijms-26-06170-f001:**
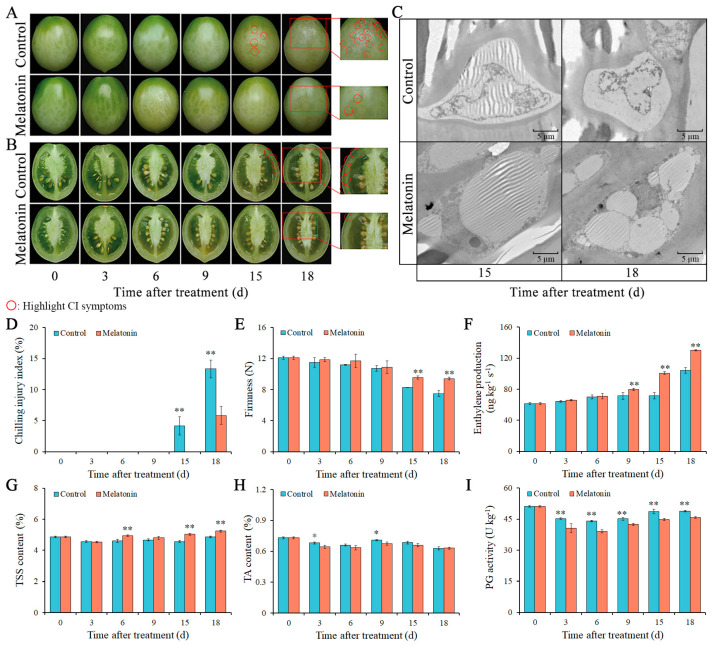
Effects of MT treatment on tomato fruit during cold storage. (**A**) CI phenotypic changes in the exterior of tomato fruit. (**B**) CI phenotypic changes in the transverse section of tomato fruit. (**C**) Variations in cell ultrastructure of tomato pericarp. Right scale bar: 5 µm. (**D**) CI index changes in tomato fruit. (**E**) Firmness of tomato fruit. (**F**) Ethylene production of tomato fruit. (**G**) TSS content of tomato fruit. (**H**) TA content of tomato fruit. (**I**) PG activity of tomato fruit. Vertical bars represent standard deviations of the mean. Asterisks indicate statistical difference of the values at *p* ≤ 0.05 (*) or *p* ≤ 0.01 (**).

**Figure 2 ijms-26-06170-f002:**
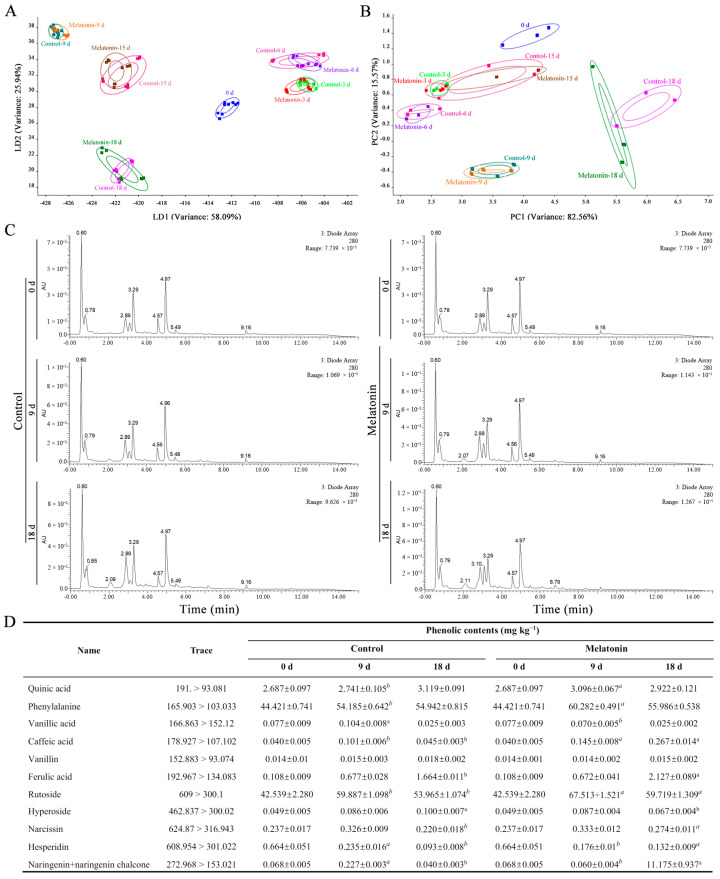
Effects of MT treatment on LDA (**A**) and PCA (**B**) in tomato fruit. (**C**) The phenolic chromatogram of the tomato fruit. (**D**) The contents of individual phenolic compounds. Data are means ± SD (standard deviation) of three replicates. The different colors represent different samples. The different letters indicate significant differences between control and MT treatment for the same storage duration (*p* ≤ 0.05).

**Figure 3 ijms-26-06170-f003:**
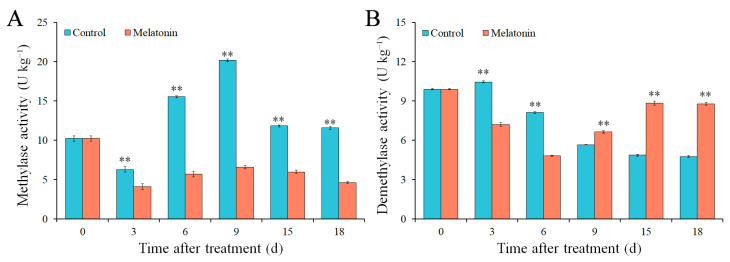
Effects of MT on methylase (**A**) and demethylase (**B**) activity of tomato fruit. Vertical bars represent standard deviations of the mean. Asterisks indicate statistical difference of the values at *p* ≤ 0.01 (**).

**Figure 4 ijms-26-06170-f004:**
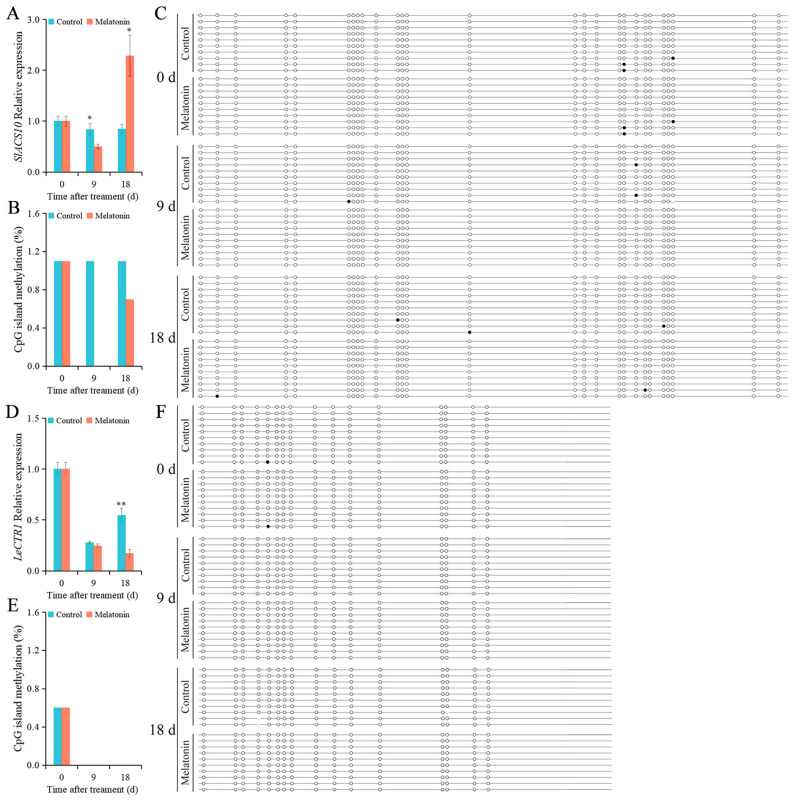
The expression and DNA methylation levels of *SlACS10* and *LeCTR1* in tomato fruit. (**A**) The relative expression level of *SlACS10*. (**B**) The DNA methylation level of CpG island of *SlACS10*. (**C**) Sequencing of DNA methylation sites of CpG island of *SlACS10*. (**D**) The relative expression level of *LeCTR1*. (**E**) The DNA methylation level of CpG island of *LeCTR1*. (**F**) Sequencing of methylation sites of CpG island of *LeCTR1*. Each line represents 1 clone, and 1 circle represents 1 CpG site. The black circle represents the methylated CG, and the white circle represents the unmethylated CG. Vertical bars represent standard deviations of the mean. Asterisks indicate statistical difference of the values at *p* ≤ 0.05 (*) or *p* ≤ 0.01 (**).

**Figure 5 ijms-26-06170-f005:**
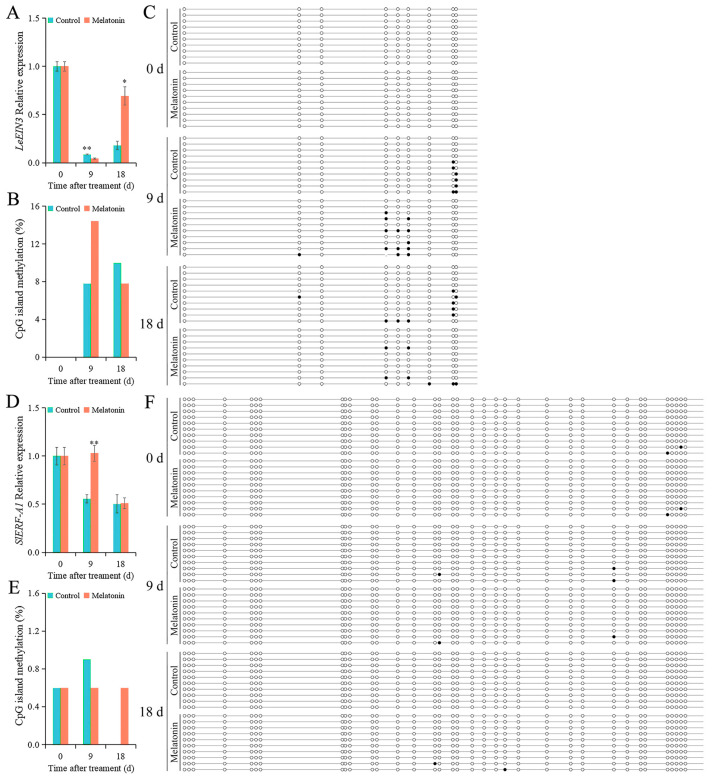
The expression and DNA methylation levels of *LeEIN3* and *SlERF-Al* in tomato fruit. (**A**) The relative expression level of *LeEIN3*. (**B**) The DNA methylation level of CpG island of *LeEIN3*. (**C**) Sequencing of DNA methylation sites of CpG island of *LeEIN3*. (**D**) The relative expression level of *SlERF-Al*. (**E**) The DNA methylation level of CpG island of *SlERF-Al*. (**F**) Sequencing of DNA methylation sites of CpG island of *SlERF-Al*. Each line represents 1 clone, and 1 circle represents 1 CpG site. The black circle represents the methylated CG, and the white circle represents the unmethylated CG. Vertical bars represent standard deviations of the mean. Asterisks indicate statistical difference of the values at *p* ≤ 0.05 (*) or *p* ≤ 0.01 (**).

**Figure 6 ijms-26-06170-f006:**
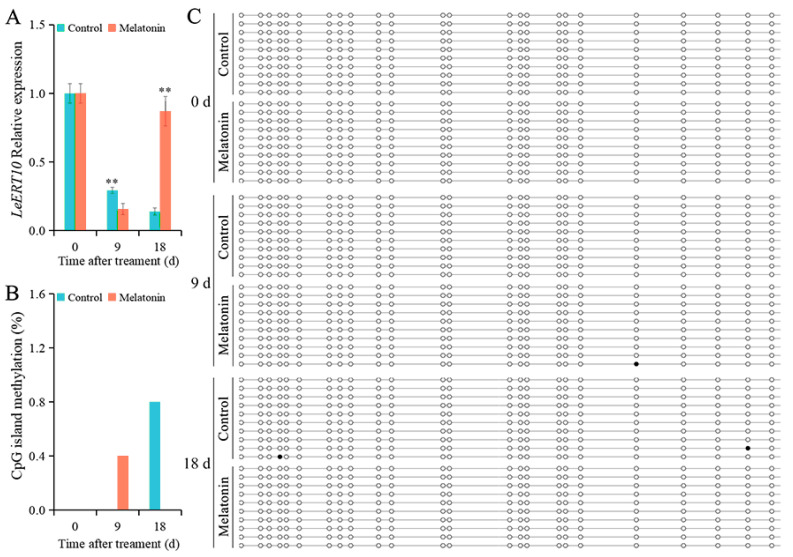
The expression and DNA methylation levels of *LeERT10* in tomato fruit. (**A**) The relative expression level of *LeERT10*. (**B**) The DNA methylation level of CpG island of *LeERT10*. (**C**) Sequencing of DNA methylation sites of CpG island of *LeERT10*. Each line represents 1 clone, and 1 circle represents 1 CpG site. The black circle represents the methylated CG, and the white circle represents the unmethylated CG. Vertical bars represent standard deviations of the mean. Asterisks indicate statistical difference of the values at *p* ≤ 0.01 (**).

**Figure 7 ijms-26-06170-f007:**
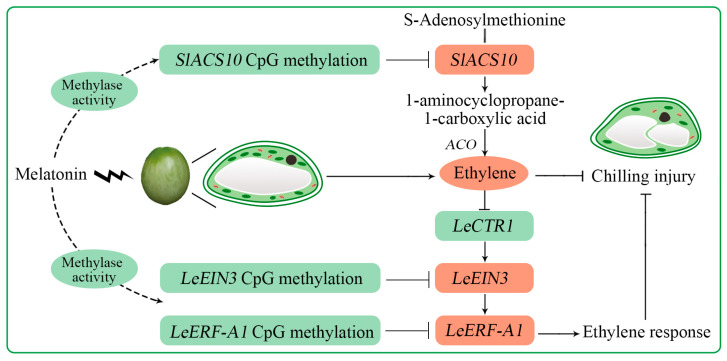
A simplified model of the regulatory mechanism of the melatonin acting while tomato fruit suffers chilling injury. The red represents upregulation, and the green represents downregulation. The arrow represents promotion, and the short line represents inhibition.

**Table 1 ijms-26-06170-t001:** Primers used in the qRT-PCR.

Gene	Primer	Sequence (5’-3’)
*GAPDH*	Forward Primer Reverse Primer	AGCCACTCAGAAGACCGTTG AGGTCAACCACGGACACATC
*SlACS10*	Forward Primer Reverse Primer	GCTCAATGCATTTGCAGTCTTG CCACAGGATTCGAGGGGTTAG
*LeCTR1*	Forward Primer Reverse Primer	GCAGCAGACGGAAGAGAGTT CTGAGCAGGAGCCCAAACA
*LeEIN3*	Forward Primer Reverse Primer	TTGATCGAAATGGCCCTGCT GGGTGGAGATAACCCCCTTCT
*SlERF-A1*	Forward Primer Reverse Primer	GGCGAAAAATGGAGCACGAG CCACGAGCAACCTTCTTCCT
*LeERT10*	Forward Primer Reverse Primer	ATTGAAGCCGCCGTACAGAA CAATCTCACCTCGAAAGCCG

**Table 2 ijms-26-06170-t002:** Primers used in the bisulfite sequencing PCR.

Gene	Primer	Sequence (5’-3’)
*SlACS10*	Forward Primer Reverse Primer	GGTTAGGTAGTTGATTGA(C/T)GTTATATT CAAATACCTAAAATTACCCAATAATT
*LeCTR1*	Forward Primer Reverse Primer	GTATTTGATTTGGATTTGATGGATT TACCAATACATCAATCACAAAATCC
*LeEIN3*	Forward Primer Reverse Primer	GTGGAGTTTAAGAAGTTGAGTATAAGT CATCATTTTCAACATATACTTCAATAT
*SlERF-A1*	Forward Primer Reverse Primer	T(C/T)GGAATTTGTGGTTTTATTAGAGA ATTACTTTATTCCAC(G/A)AACAACCTTC
*LeERT10*	Forward Primer Reverse Primer	TGCAATCTCTATGTGATGAAATCAACT TCATAACTAGTCATTTCAAGTTCAAC

## Data Availability

Data are contained within the article or Appendix A.

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
