# Peer review of "DNA Methylation Profile Changes in CpG Islands of Ethylene-Signaling Genes Regulated by Melatonin Were Involved in Alleviating Chilling Injury of Postharvest Tomato Fruit"

_ijms, 2025, doi:10.3390/ijms26136170_

Round 1
Reviewer 1 Report
Comments and Suggestions for Authors
The manuscript entitled “DNA methylation profile changes of CpG islands of ethylene signaling genes regulated by melatonin were involved in alleviating chilling injury of postharvest tomato fruit” by Jingrui Yan et al. reports interesting observations that may link DNA methylation to ethylene, melatonin and chilling injury. The manuscript is clearly and well written and in principle, can make a worthy contribution to post-harvest chilling biology in tomato if properly revised.
There are in my opinion 4 points that need addressing:
1) I do not see any information about the procedure to solublilize melatonin. Melatonin looks like a substance that is highly polar and presumably not soluble in water. So, how did the authors put it into an aqueous solution. Related to this point is the question of the adequate control. If indeed the authors used any organic solvent, this solvent needs to be present in the control - rather than just water as the authors claim.
2) It seems to me that there is no proper reference made to how melatonin functions in plants. Is the action of melatonin dependent on particular (dedicated ) receptors ? If the action is mediated by a dedicated receptor, then the question arises of where the downstream signal of the receptor interacts with the ethylene pathway. I propose that the authors have a closer look at a recent review as some sort of a starting point: Front. Plant Sci., 14 February 2023, Sec. Plant Abiotic Stress, Volume 14 - 2023 https://doi.org/10.3389/fpls.2023.1108507
3) Related to point 2, my next comment relates to the model which I think is not a model in the strict sense of the word but rather a graphic summary of the observations. We do not learn anything about possible mechanisms of action and any causal relationship between components.
4) The authors must tell us why they tested the methylation status of the particular genes they have chosen (e.g. SIACS10 etc.). If the reason is, as I suspect, a link to the ethylene pathway, then there must be a control of the methylation status in response to melatonin in a “household” gene. Only then can there be some assurance that melatonin is not a general and unspecific modulator of DNA methylation.
In summary, if the authors embark on a serious quest to address these points, this manuscript has a good chance to make a contribution to our understanding of the molecular processes that govern chilling injury.
Author Response
Reponses to the comments of reviewer #1:
- I do not see any information about the procedure to solublilize melatonin. Melatonin looks like a substance that is highly polar and presumably not soluble in water. So, how did the authors put it into an aqueous solution. Related to this point is the question of the adequate control. If indeed the authors used any organic solvent, this solvent needs to be present in the control - rather than just water as the authors claim.
Response: Thanks for your suggestion. The water solubility of melatonin (CAS: 73-31-4, molar mass: 232.28 g mol-1) was 434.4 mg L-1 at room temperature (25 oC). In this study, we used the 500 µM melatonin solution to immerse tomato fruit at 25 oC for 10 min. Specifically, the 116.14 mg of melatonin was weighed, and then dissolve in 1 L of water, this concentration was less than its solubility in water. Previous studies demonstrated that postharvest application of melatonin (1,000 µM) could be used commercially to attenuating Aspergillus decay, suppressing AB1 accumulation, and keeping the quality of fresh pistachio fruits during postharvest life (Jannatizadeh et al., 2021). The optimal melatonin treatment (0.5 mM) can maintain postharvest organoleptic quality of Hami melon fruit by increasing its antioxidant activity and inhibiting reactive oxygen species production (Wang et al., 2023). The concentrations of melatonin used in these studies were higher than or equal to our study, while they only dissolve melatonin in water.
Jannatizadeh, A.; Aminian-Dehkordi, R.; Razavi, F. Effect of exogenous melatonin treatment on Aspergillus decay, aflatoxin B1 accumulation and nutritional quality of fresh “Akbari” pistachio fruit. J. Food Process Preserv. 2021, 45, e15518. https://doi.org/10.1111/jfpp.15518.
Wang, Y.; Guo, M.; Zhang, W.; Gao, Y.; Ma, X.; Cheng, S.; Chen, G. Exogenous melatonin activates the antioxidant system and maintains postharvest organoleptic quality in Hami melon (Cucumis. melo var. inodorus Jacq.). Front. Plant Sci. 2023, 14, 1274939. https://doi.org/10.3389/fpls.2023.1274939
- It seems to me that there is no proper reference made to how melatonin functions in plants. Is the action of melatonin dependent on particular (dedicated) receptors? If the action is mediated by a dedicated receptor, then the question arises of where the downstream signal of the receptor interacts with the ethylene pathway. I propose that the authors have a closer look at a recent review as some sort of a starting point: Front. Plant Sci., 14 February 2023, Sec. Plant Abiotic Stress, Volume 14 - 2023 https://doi.org/10.3389/fpls.2023.1108507.
Response: Thanks for your suggestion. According to your suggestion, we have been read carefully this review and added related description about melatonin (Lines 49-60).
- Related to point 2, my next comment relates to the model which I think is not a model in the strict sense of the word but rather a graphic summary of the observations. We do not learn anything about possible mechanisms of action and any causal relationship between components.
Response: Thanks for your suggestion. According to your suggestion, we have modified the Figure 7 and added the relevant description.
- The authors must tell us why they tested the methylation status of the particular genes they have chosen (e.g. SIACS10 etc.). If the reason is, as I suspect, a link to the ethylene pathway, then there must be a control of the methylation status in response to melatonin in a “household” gene. Only then can there be some assurance that melatonin is not a general and unspecific modulator of DNA methylation.
Response: Thanks for your suggestion. In the present study, we screened more known genes related to ethylene synthesis and signaling transduction in tomato fruit, including ACS10, CTR1, EIN3, ERF-A1, ERT10, ACS2, ACS4, ACO1, ACO2, EIN2, ETR3, EIL3, etc. However, ACS2, ACS4, ACO1, ACO2, ETR3 and EIL3 genes have no CpG islands, and could not be used in the detection. EIN2 and other genes have CpG islands, but contain high content of CG, and there are no methods to sequence them in the current technology. In addition, since green fruit were used in this study, some genes could not be detectable because they are mainly expressed during ripening. Hence, in the present study, only the five genes of SlACS10, LeCTR1, LeEIN3, SlERF-A1 and LeERT10 were selected.
Reviewer 2 Report
Comments and Suggestions for Authors
This study provides novel evidence that melatonin alleviates chilling injury in postharvest tomato fruit during cold storage by modulating DNA methylation patterns within CpG islands of key ethylene signaling genes (SlACS10, LeCTR1, LeEIN3, SlERF-A1, LeERT10), consequently altering their expression (inhibiting LeCTR1 while enhancing SlACS10, LeEIN3, SlERF-A1), which appears to activate ethylene signaling pathways and improve fruit quality attributes (increased soluble solids, firmness, and ethylene production; decreased acidity and polygalacturonase activity) under chilling stress.
General comments and queries:
- Melatonin increased ethylene production (Fig. 1F) and expression of SlACS10 and LeEIN3/SlERF-A1, but inhibited LeCTR1. However, no direct evidence confirms ethylene signaling activation (e.g., ethylene-responsive reporter genes). How was "activation" of ethylene signaling conclusively demonstrated?
- Higher phenolic compounds in melatonin-treated fruit (Fig. 2D) may contribute to chilling injury tolerance via antioxidant effects. Were these compounds functionally linked to the ethylene-methylation axis, or are they parallel, independent effects of melatonin?
- Figure 4: For LeCTR1, expression decreased significantly with melatonin, but no CpG methylation changes were detected. Could you please explain this difference? Could melatonin regulate LeCTR1 via non-epigenetic mechanisms?
- Electronic nose data showed altered volatile profiles in melatonin fruit but was disconnected from the ethylene/methylation narrative. What is the hypothesized role of volatiles in chilling injury alleviation?
- Conclusion: It is a very short conclusion. Please mention the salient features/outcome of the study here and suggest any future studies related to the subject matter.
I have suggested some changes in the attached PDF, which are self-explanatory.

Author Response
Reponses to the comments of reviewer #2:
- Melatonin increased ethylene production (Fig. 1F) and expression of SlACS10 and LeEIN3/SlERF-A1, but inhibited LeCTR1. However, no direct evidence confirms ethylene signaling activation (e.g., ethylene-responsive reporter genes). How was "activation" of ethylene signaling conclusively demonstrated?
Response: Thanks for your suggestion. The ethylene signal transduction pathway is responsible for the perception and signaling of the hormone. In the absence of ethylene, the receptors and CTR1 interact to suppress the downstream signaling component EIN2 by keeping it in its phosphorylated. In the presence of ethylene, ethylene binding to the receptors leads to a conformational change, that affects the interaction with CTR1 and prevents the phosphorylation of EIN2. Furthermore, upon ethylene binding the receptors are degraded by the 26S proteasome dependent pathway. As a consequence, EIN2 is no longer being phosphorylated which triggers the cleavage of its C-terminal domain (CEND), that migrates to the nucleus. In the nucleus, the EIN2-CEND prevents the degradation of EIN3. EIN3 can activate the expression of ERF. Subsequently, the ERFs control a large number of ethylene-responsive genes inducing various ethylene-mediated responses (Mata Clara et al., 2021). Therefore, the activation of ethylene signal is a series of complicated chain reactions. In this study, we presume that melatonin activates ethylene signaling pathway according to the variation of ethylene production, the level of expression and methylation of LeEIN3 and SlERF-A1 gene. According to your suggestion, we found that the description about "activation" is not rigorous enough, we have been modified it in the whole manuscript (Lines 24, 296, 319).
Mata Clara, I.; Magpantay, J.; Hertog Maarten, L.A.T.M.; Van de Poel, B.; Nicolai Bart, M. Expression and protein levels of ethylene receptors, CTRs and EIN2 during tomato fruit ripening as affected by 1-MCP. Postharvest Biol. Tec. 2021, 179, 111573. https://doi.org/10.1016/j.postharvbio.2021.111573.
- Higher phenolic compounds in melatonin-treated fruit (Fig. 2D) may contribute to chilling injury tolerance via antioxidant effects. Were these compounds functionally linked to the ethylene-methylation axis, or are they parallel, independent effects of melatonin?
Response: Thanks for your suggestion. Previous studies demonstrated that exogenous melatonin treatment effectively promoted phenolic compounds synthesis by enhancing the accumulation of the family of aromatic amino acids, in particular, the Phe that precursors of phenolic compound, and which up-regulated mRNA levels of genes involved in phenolic biosynthesis (PaPAL, PaC4H, Pa4CL, PaCHS, PaF3H, PaF3’H, PaDFR, PaANS, PaUFGH) (Pang et al., 2023). Moreover, ethylene treatment provided higher concentrations of phenolic compounds in persimmon fruit for astringency removal (Ancillotti et al., 2019). In this study, the changes of phenolic compounds might be influent by dual effects of melatonin and ethylene.
Pang, L.L.; Chen, L.; Jiang, Y.Q.; Zhou, C.; Liang, F.H.; Duan, L.H. Role of exogenous melatonin in quality maintenance of sweet cherry: Elaboration in links between phenolic and amino acid metabolism. Food Biosci. 2023, 56, 103223. https://doi.org/10.1016/j.fbio.2023.103223.
Ancillotti, C.; Caprini, C.; Scordo, C.; Renai, L.; Giordani, E.; Orlandini, S.; Furlanetto, S.; Bubba, M. D. Phenolic compounds in Rojo Brillante and Kaki Tipo persimmons at commercial harvest and in response to CO2 and ethylene treatments for astringency removal. LWT. 2019, 100, 99-105. https://doi.org/10.1016/j.lwt.2018.10.031.
- Figure 4: For LeCTR1, expression decreased significantly with melatonin, but no CpG methylation changes were detected. Could you please explain this difference? Could melatonin regulate LeCTR1 via non-epigenetic mechanisms?
Response: Thanks for your suggestion. Previous studies demonstrated that exogenous melatonin treatment can promote ethylene content and stimulate ripening of tomato fruit (Sun et al., 2015). The ethylene was combined with ethylene receptors (ETR) to inhibit the activity of constitutive triple response 1 (CTR1) (Mata Clara et al., 2021). The similar result was observed in our study. In this study, the CpG methylation changes of LeCTR1 were not detected differences between control and melatonin treatment, while the expression of LeCTR1 decreased significantly in melatonin treated fruit at 18 d, which might be melatonin treatment significantly increased the ethylene production to inhibit the relative expression of LeCTR1, rather than epigenetic mechanisms.
Sun, Q.; Zhang, N.; Wang, J.; Zhang, H.; Li, D.; Shi, J.; Li, R.; Weeda, S.; Zhao, B.; Ren, S.; Guo, Y. D. Melatonin promotes ripening and improves quality of tomato fruit during postharvest life. J. Exp. Bot. 2015, 66, 657-668. https://doi.org/10.1093/jxb/eru332.
Mata Clara, I.; Magpantay, J.; Hertog Maarten, L.A.T.M.; Van de Poel, B.; Nicolai Bart, M. Expression and protein levels of ethylene receptors, CTRs and EIN2 during tomato fruit ripening as affected by 1-MCP. Postharvest Biol. Tec. 2021, 179, 111573. https://doi.org/10.1016/j.postharvbio.2021.111573.
- Electronic nose data showed altered volatile profiles in melatonin fruit but was disconnected from the ethylene/methylation narrative. What is the hypothesized role of volatiles in chilling injury alleviation?
Response: Thanks for your suggestion. Aroma is an important commercial trait that determines fruit quality, which influence the acceptance of consumers to fruit (Chen et al., 2023). Previous studies demonstrated that the tomato fruit tomato fruit was recommended storage temperature of 10 °C for delaying postharvest senescence, while which was able to result in a significant negative effect on the aromatic flavor of tomato. Compared with that at 10 °C alone, the 1-MCP treatment under 10 oC not only delay the shelf life of tomato fruit, but also maintain the flavor quality of tomato fruit (Zou et al., 2018). In our study, the melatonin treatment effectively not only alleviated chilling injury, but also maintained the volatile aroma during low temperature storage, which both increased the commercial trait of tomato fruit.
Chen, Y.Y.; Wu, X.; Wang, X.H.; Yuan, Y.B.; Qi, K.J.; Zhang, S.L.; Yin, H. PusALDH1 gene confers high levels of volatile aroma accumulation in both pear and tomato fruits. J. Plant Physiol. 2023, 290, 154101. https://doi.org/10.1016/j.jplph.2023.154101.
Zou, J.; Chen, J.; Tang, N.; Gao, Y.Q.; Hong, M.S.; Wei, W.; Cao, H.H.; Jian, W.; Li, N.; Deng, W.; Li. Z.G. Transcriptome analysis of aroma volatile metabolism change in tomato (Solanum lycopersicum) fruit under different storage temperatures and 1-MCP treatment. Postharvest Biol. Tec. 2018, 135, 57-67. http://dx.doi.org/10.1016/j.postharvbio.2017.08.017.
- Conclusion: It is a very short conclusion. Please mention the salient features/outcome of the study here and suggest any future studies related to the subject matter.
Response: Thanks for your suggestion. According to your suggestion, we have been added related description in the conclusion (Lines 442-450).
- I have suggested some changes in the attached PDF, which are self-explanatory.
Response: Thanks for your suggestion. According to your suggestion, we have been reasonably modified manuscript.
Round 2
Reviewer 1 Report
Comments and Suggestions for Authors
The manuscript is acceptable in its current form.